# The Improved Effects of a Multidisciplinary Team on the Survival of Breast Cancer Patients: Experiences from China

**DOI:** 10.3390/ijerph17010277

**Published:** 2019-12-31

**Authors:** Jianlong Lu, Yan Jiang, Mengcen Qian, Lilang Lv, Xiaohua Ying

**Affiliations:** 1NHC Key Laboratory of Health Technology Assessment (Fudan University), School of Public Health, Fudan University, Dongan Road 130, Shanghai 200032, China; 17111020025@fudan.edu.cn (J.L.); qianmengcen@fudan.edu.cn (M.Q.); 2Shanghai Cancer Center, Fudan University, Dongan Road 270, Shanghai 200032, China; sandyjy@126.com (Y.J.); lvlanglang@hotmail.com (L.L.)

**Keywords:** multidisciplinary team meeting, breast cancer, propensity score matching, survival analysis

## Abstract

This study aimed to explore whether different multidisciplinary team (MDT) organizations have different effects on the survival of breast cancer patients. A total of 16354 patients undergoing breast cancer surgery during the period 2006–2016 at the Fudan University Shanghai Cancer Center were retrospectively extracted. Patients treated by MDT were divided into a well-organized group and a disorganized group based on their organized MDT, professional attendance, style of data and information delivery, and the length of discussion time for each patient. Other patients, who were not treated by MDT, were placed in a non-MDT group as a comparator group. Each MDT patient was matched with a non-MDT patient, using propensity score matching to reduce selection bias. The Cox regression model was used to examine the difference in effects between groups. We found that the five-year survival rate of the well-organized MDT group was 15.6% higher than the non-MDT group. However, five-year survival rate of the disorganized MDT group was 19.9% lower than that of the non-MDT group. Patients in the well-organized MDT group had a longer survival time than patients in the non-MDT group (HR = 0.4), while the disorganized MDT group had a worse survival rate than the non-MDT group (HR = 2.8) based on the Cox model result. However, our findings indicate that a well-organized MDT may improve the survival rate of patients with breast cancer in China.

## 1. Introduction

Multidisciplinary teams (MDTs) are composed of healthcare professionals (including surgeons, oncologists, radiologists, pathologists, pathologists, and specialist nurses), and aim to reach a consensus on the diagnosis and treatment of patients, based on scientific and experiential evidence [1]. A coordinator is responsible for organizing the MDT meetings [2]. MDTs make decisions regarding diagnosis and treatment programs through MDT meetings [3].

MDT meetings have been performed as part of cancer care services in Europe, the United States, and Australia [4,5]. They have been considered to be the gold standard of cancer care [6]. Many guidebooks have been published to promote the effectiveness of MDT meetings in developed countries, which emphasize certain elements, such as the composition of MDTs, meeting attendance, teamwork, information exchange, and the equipment required for meetings [2,3]. China has a large number of cancer patients, making it difficult to provide every patient with multidisciplinary consultation [7]. Indeed, few MDTs have been organized since 2005, except in some large cities, such as Beijing, Shanghai, and Guangzhou [7]. There was no substantial change until 2015, at which point many cancer treatment consensuses began to cover suggestions of treatment by MDTs [8,9,10,11,12,13,14,15,16].

Fudan University Shanghai Cancer Center (FUSCC) was one of the first hospitals to start MDT meetings in China, and has accumulated much experience in building effective MDTs. During the period 2005–2010, many adjustments were made to improve the effectiveness of MDTs. First, in 2007, the FUSCC team hired a secretary to coordinate and administer the process of MDTs. Second, a pathologist joined the team in 2008, making the team more complete. Third, they started to use a new meeting room and equipment in 2009 to improve searches and discuss patient information in real time. Fourth, in 2010, they limited the total number of patients in each meeting to ensure there was enough discussion time for each patient. Patients’ information was also shared among MDT members before the meeting. Numerous studies have demonstrated that these adjustments are the core requirements for a well-organized MDT [1,2,17]. We thus divided the MDT group into two subgroups based on whether the MDT’s organization incorporates these core elements. The subgroup consisting of patients treated during the period 2006–2010 was defined as the “Disorganized MDT” group, and the subgroup consisting of patients treated during the period 2011–2016 was defined as the “Well-organized MDT” group. In other words, a well-organized MDT has a more intact team, more appropriate infrastructure for meetings, and better organization of meetings than a disorganized MDT. Table 1 shows the main differences between the two groups. The control group (non-MDT) included patients who did not receive MDT treatment during the period 2006–2016.

Many studies have assessed the effect of MDT meetings [18,19,20,21] and have suggested that patients treated by MDT had a longer survival time than other patients. The effectiveness of an MDT is associated with its organization, process, and the disease being treated. These vary across MDT organizations and countries. According to the theory of Group Decision Making, different MDT organizations can make different decisions on the same medical issue. However, previous studies might have overlooked these influencing factors, so it is still difficult to distinguish which type of MDT organization is more effective. Besides, there was no evidence-based research previously carried out on the effectiveness of MDTs in China.

Using data from FUSCC, this study aimed to examine whether breast cancer patients treated by a well-organized MDT might experience a prolonged survival time compared with patients who did not receive MDT treatment, or received disorganized MDT treatment.

## 2. Materials and Methods

### 2.1. Participants

FUSCC is one of the most well-known cancer centers in China. In this study, patients who received breast cancer surgeries in FUSCC between the period 2006–2016 were studied. However, patients without basic information, such as TNM or follow-up records, were excluded. Patients were grouped into an MDT group or an N-MDT group based on whether they were treated by an MDT or not (Figure 1).

### 2.2. Study Design

FUSCC optimized the membership and organization of the MDT team during the period 2005–2011. We thus defined patients treated by an MDT during the period 2006–2010 as the disorganized MDT group and patients treated by an MDT during the period 2011–2016 as the well-organized MDT group. This study is an observational retrospective cohort study. The study design is shown in Figure 2. The protocol has been approved by the Fudan University Shanghai Cancer Center Institutional Review Board (SCCIRB-1812195-18).

### 2.3. Variables

Mortality information was obtained from the hospital’s follow-up system. The survival time was defined as the time period between the date of surgery and the date of death for truncated patients, or the last follow-up time for censored patients. The Charlson comorbidity index (CCI) was used to represent comorbid diseases. The tumor node metastasis (TNM), immunohistochemical index estrogen receptor (ER), and epidermal growth factor receptor 2 (HER2), were included to represent disease severity.

### 2.4. Data Analysis

A propensity score matching (PSM) with a ratio of 1:1 was used to match the patients in the MDT group and N-MDT group based on the time of surgery, Charlson comorbidity index (CCI), tumor node metastasis (TNM), bilateral incidence, and the immunohistochemical indexes ER and HER2. A Cox proportional hazards model, with the survival data of patients treated after 2011, was used to quantify the effect of a well-organized MDT. A Schoenfeld residual test was used to test the proportional hazards assumption. This model was also used to analyze the hazard ratio of MDT, age, the Charlson comorbidity index, TNM, bilateral incidence, ER, HER2, and the effect of the difference between MDTs before and after 2011. To testify that there was a difference between a well-organized MDT and a disorganized MDT, an interaction term (MDT*TIME) was introduced in the Cox regression, using the matched dataset from 2006 to 2016. If patients were treated before 2011, the index “TIME” was “0”, otherwise the index “TIME” was “1”. Kaplan-Meier survival curves were conducted to show differences in survival between groups. The 1-, 3- and 5-year survival rate, and median survival time, were calculated for all groups. Inter-group differences were analyzed using the two-sample Wilcoxon rank-sum (Mann-Whitney) test, chi-square, or log-rank test. Data were analyzed using STATA 12.0.

## 3. Results

### 3.1. Basic Characteristics of the Pre-PSM and Post-PSM Samples

We extracted 16,354 eligible patients from the database, including 299 MDT patients and 16,055 N-MDT patients (Table 2). The average age was 50.2 ± 13.0 years for MDT patients, and 51.9 ± 11.2 years for N-MDT patients. Patients in the MDT group had a higher proportion of Stage 3 tumors compared to patients in the N-MDT group, at 50.8% versus 12.7%, respectively. Nearly 90% of patients in the MDT group were unilateral, compared to 98.4% for the N-MDT group. The percentage of positive ER and negative HER2 for the N-MDT were 58.2% and 59.5%, while the equivalent percentages were 44.5% and 51.2% for the MDT group. Overall, the MDT group showed a significantly higher Charlson comorbidity index result (*p* = 0.016), a significantly higher stage 3 and 4 proportion result (*p* < 0.001), a significantly higher negative ER proportion result (*p* < 0.001), and a higher proportion of positive HER2 (*p* = 0.034) than the N-MDT group.

After PSM, the sizes of the MDT and N-MDT groups were both reduced to 218 patients, with the inter-group differences in the Charlson comorbidity index, TNM, bilateral incidence, and immunohistochemical index ER or HER2 disappearing (Table 3).

### 3.2. Overall Survival Differences between MDT Group and N-MDT Group During 2006--2016

Table 4 shows that the 1-year, 3-year, 5-year survival rate, and median survival time in the MDT group were 98.5%, 81.6%, 65.6%, and 1131 days, and the equivalents for the N-MDT group were 97.6%, 77.4%, 72.8%, and 946 days, respectively. However, there was no statistically significant evidence that the survival rates and times were higher.

In Panel B after 2011, the 1-year, 3-year, 5-year survival rate, and median survival time in the MDT group were 98.1%, 84.1%, 78.8%, and 790 days, and were 95.4%, 67.9%, 63.3%, and 647 days in the N-MDT group. In particular, there were significant differences in the 3- and 5-year survival rate and the median survival time (*p* = 0.004 for 3 years, *p* = 0.007 for 5 years, *p* = 0.043 for median survival time).

### 3.3. Differential Effects of MDT before and after the Year of 2011

Table 5 shows that the effect of MDT varied across different “TIME” inputs. When TIME = 0, having an MDT treatment was not beneficial to the survival rate (HR = 2.8, *p* < 0.001). However, when TIME = 1, which was assessed using the sample of Panel B, having an MDT treatment became a protective factor for the survival of patients (HR = 0.4, *p* = 0.014, Appendix A). The Cox regression met the proportional hazards assumption (*p* = 0.102 for Cox regression with interaction term, *p* = 0.226 for Cox regression of Panel B).

Figure 3. shows the survival curves of the disorganized MDT (before 2011), well-organized MDT group (after 2011), and the corresponding N-MDT group, respectively. Well-organized MDT patients have a better survival curve than the N-MDT group in Panel B (log-rank test, *p* = 0.013), while the disorganized MDT patients show the opposite result in Panel A ((log-rank test, *p* = 0.001)).

## 4. Discussion

### 4.1. The Explanation of the Results

The results suggest that having an MDT had a more positive effect between 2011–2016 than during 2006–2010. One explanation for this is that the MDTs were more organized during 2011–2016. This means that having an MDT secretary, adding a pathologist, having a suitable infrastructure, providing enough discussion time, and being well prepared may have a positive effect on the effectiveness of MDTs. Certain reasons can explain this. First, the secretary and the pathologist play important roles in the team. The secretary can make the meeting more efficient, and the pathologist can directly influence the treatment program. Second, a suitable meeting room and equipment can improve the quality of discussion and help obtain more accurate information. Third, regulations on discussion time and delivering patient information in advance can help the team make more precise judgments. Some studies suggest these steps as core measures for making efficient MDTs. However, the changing of those factors in our study was also the feature of the time period instead of a tight intervention portfolio, and further study was therefore needed to figure out the relationship between those factors and the outcome of the MDT.

This study suggests that well-organized MDTs benefit breast cancer patients who are suitable for surgery, and the organization of MDTs can influence the size of their effects. Our findings are consistent with previous studies on breast cancer [18,22,23,24,25]. The patients discussed by a well-organized MDT had a 15.6% higher survival rate at five years than those who were not discussed by an MDT. This result was similar to that of Eileen M Kesson’s study (11%–18% lower mortality rates) [18]. The influence of an MDT was mainly shown in their treatment decisions [26]. However, the disorganized MDT, which was before 2011, showed no such effects. The lack of regulations on discussion time, engagement, and equipment influenced the final decision of the MDT. Therefore, making sure the MDT runs efficiently is more important than setting up an MDT. In our study, the 5-year survival rate of non-MDT patients was higher than those discussed by disorganized MDTs, but lower than those discussed by well-organized MDTs. The result also suggest that the survival of patients discussed by a disorganized MDT was worse than for N-MDT patients. The reasons for these results were that the severity of disease in MDT groups, such as the TNM stage (53% in stage 3) and the average of CCI (0.26), was higher than in N-MDTs (51% in stage 3, 0.2 for average CCI). People with more severe situations, such as a bad basic status or dangerous locations, are more likely to be recommended to receive MDT treatments. So, the worse survival situation may be caused by the disease severity instead of MDT.

### 4.2. Strengths and Limitations of This Study

This study has several strengths. First, it is the first study quantitatively showing the different effects among different MDT organizations. Second, it is one of the few studies focusing on the effect of MDTs in China.

This study has several limitations. First, the patients in this study were collected from the Hospital Information System of one hospital, resulting in a small sample size and missing data. This may result in selection bias. Second, this study was limited to these indicators. This may influence the results due to a lack of sufficient covariates. For example, the basic status of patients, tumor imaging information, and tumor location are the main factors doctors use to recommend an MDT treatment, which were hard to acquire for our study. Third, while we included the year of surgery as a PSM factor in the study design to avoid the influence of the treatment progress, the confounding effects over the 10 years may not have been completely eliminated, so, we cannot totally distinguish the effect of MDT from the effect of the time period. Finally, we separated the MDT into two periods based on the year 2011. However, the adjustment process of MDT organization took nearly 5 years, and it was a process occurring throughout a period instead of during a point in time.

## 5. Conclusions

We found that breast cancer patients discussed by a well-organized MDT had a 15.6% higher survival rate after five years than those who were not discussed by MDTs. However, five-year survival rate of disorganized MDT patients was 19.9% lower than that of non-MDT patients. This result suggests that, after an efficient multidisciplinary team discussion, patients eligible for breast cancer surgery have a prolonged survival rate compared to patients who did not receive MDT discussions. A number of factors may influence the effect of an MDT, including having an MDT secretary, adding a pathologist, having suitable infrastructure, providing enough discussion time, and being well prepared. Future research is needed to clarify these factors’ direct effects on the survival of patients.

## Figures and Tables

**Figure 1 ijerph-17-00277-f001:**
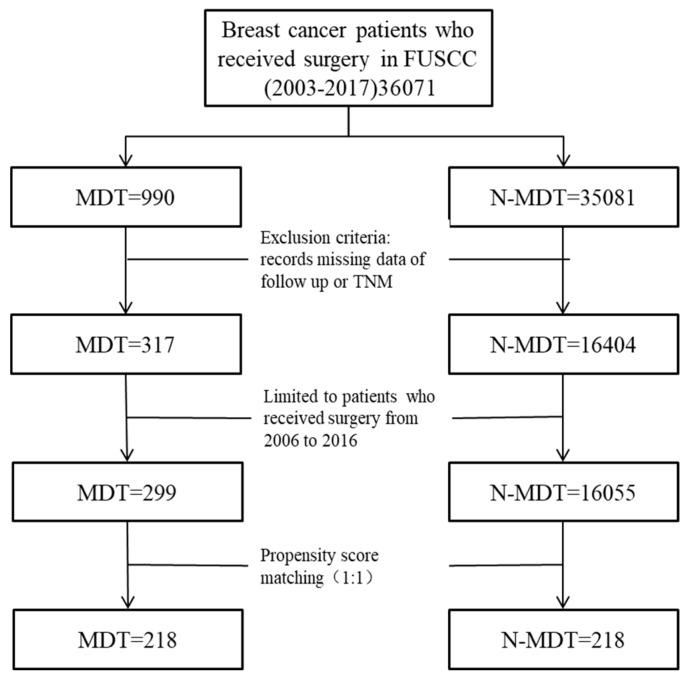
Study flow chart. MDT: Multidisciplinary teams; N-MDT: without multidisciplinary teams’ treatment.

**Figure 2 ijerph-17-00277-f002:**
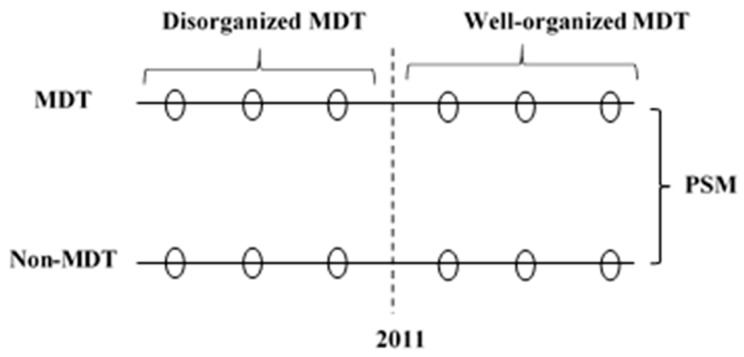
The study design. MDT: Multidisciplinary teams; N-MDT: without multidisciplinary teams’ treatment; PSM: propensity score matching.

**Figure 3 ijerph-17-00277-f003:**
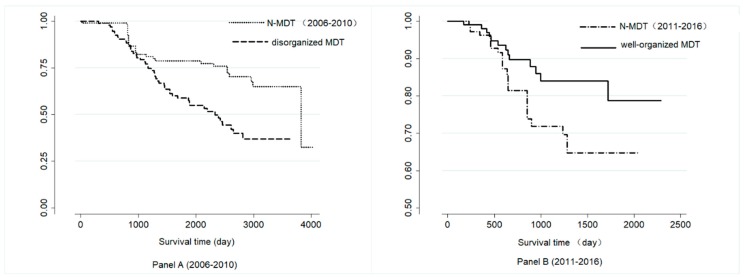
Survival curves of the different groups in Panel A and Panel B. MDT: Multidisciplinary teams, N-MDT: without multidisciplinary teams’ treatment; Panel A: patients who were treated during 2006–2010 after PSM; Panel B: patients who were treated during 2011–2016 after PSM.

**Table 1 ijerph-17-00277-t001:** The main organizational differences of MDT before and after 2011.

Determinants	Before 2011	After 2011
	(Disorganized MDT)	(Well-Organized MDT)
Organization	Chairman	Secretary
Attendance	Surgeons, physicians, Imaging doctors	Surgeons, physicians, Imaging doctors, Pathology doctor
Information delivery	At the meeting	Before the meeting
Number of patients	Unlimited	About four patients
Discussion time per patient	5–10 min	20–30 min
Patient data	Photographic, paper	Electronic

MDT: Multidisciplinary teams.

**Table 2 ijerph-17-00277-t002:** Basic characteristics of pre-PSM of the sample (2006–2016).

Variables	MDT				N-MDT				MDT vs. N-MDT *p* Value ^γ^
	Before 2011 (*n* = 135)	After 2011 (*n* = 164)	Total (*n* = 299)	*p* Value ^α^	Before 2011 (*n* = 5332)	After 2011 (*n* = 10,723)	Total (*n* = 16,055)	*p* Value ^β^	
Age,yr									
mean (SD)	48.8 (11.9)	51.5 (13.8)	50.2 (12.9)	0.059	52.3 (11.2)	51.7 (11.2)	51.9 (11.2)	0.013	0.031
Charlson comorbidity index									
mean (SD)	0.2 (0.5)	0.4 (0.6)	0.3 (0.5)	0.000	0.2 (0.5)	0.2 (0.5)	0.2 (0.5)	0.337	0.016
TNM									
0	0.0%	0.0%	0.0%	0.442	1.4%	2.2%	1.9%	< 0.001	< 0.001
1	4.4%	5.5%	5.0%		35.8%	37.9%	37.2%		
2	32.6%	32.3%	32.4%		47.4%	47.6%	47.5%		
3	56.3%	46.3%	50.8%		14.8%	11.7%	12.7%		
4	6.7%	15.9%	11.7%		0.6%	0.7%	0.7%		
Bilateral incidence									
unilateral	91.9%	88.4%	90.0%	0.325	98.2%	98.5%	98.4%	0.153	< 0.001
bilateral	8.2%	11.6%	10.0%		1.8%	1.5%	1.6%		
ER									
positive	45.2%	43.9%	44.5%	0.894	60.7%	57.0%	58.2%	0.060	< 0.001
negative	37.0%	34.8%	35.8%		21.3%	21.6%	21.5%		
HER2									
positive	18.5%	26.2%	22.7%	0.091	14.8%	21.7%	19.4%	< 0.001	0.034
negative	55.6%	47.6%	51.2%		66.0%	56.3%	59.5%		

PSM: propensity score matching; TNM: tumor node metastasis; ER: estrogen receptor; HER2: epidermal growth factor receptor 2; SD: standard deviation; *p* value ^α^: the probability of the hypothesis that the difference between before and after 2011 in MDT was caused by sampling error; *p* value ^β^: the probability of the hypothesis that the difference between before and after 2011 in N-MDT was caused by sampling error; *p* value ^γ^: the probability of the hypothesis that the difference between the MDT group and the N-MDT group was caused by sampling error; the *p* value comes from the *t*-test, Wilcoxon rank-sum (Mann-Whitney) test, chi-square.

**Table 3 ijerph-17-00277-t003:** Basic characteristics of post-PSM of the sample (2006-–2016).

Variables	MDT				N-MDT				
	Before 2011 (*n* = 98)	After 2011 (*n* = 120)	Total (*n* = 218)	*p* Value ^α^	Before 2011 (*n* = 101)	After 2011 (*n* = 117)	Total (*n* = 218)	*p* Value ^β^	MDT vs. N-MDT *p* Value ^γ^
Age,yr									
mean (SD)	48.9 (11.5)	51.4 (13.2)	50.3 (12.5)	0.196	52.8 (10.2)	53.5 (11.1)	53.2 (10.7)	0.482	0.010
Charlson comorbidity index									
mean (SD)	0.2 (0.4)	0.4 (0.6)	0.3 (0.5)	0.003	0.1 (0.4)	0.2 (0.6)	0.2 (0.5)	0.009	0.399
TNM									
0	0.0%	0.0%	0.0%	0.138	0.0%	0.0%	0.0%	0.141	0.969
1	5.1%	4.2%	4.6%		5.0%	4.3%	4.6%		
2	32.7%	31.7%	32.1%		33.7%	32.5%	33.0%		
3	59.2%	47.5%	52.8%		57.4%	45.3%	50.9%		
4	3.1%	16.7%	10.6%		4.0%	18.0%	11.5%		
Bilateral incidence									
unilateral	91.8%	89.2%	90.4%	0.506	90.1%	93.2%	91.7%	0.413	0.732
bilateral	8.2%	10.8%	9.6%		9.9%	6.8%	8.3%		
ER									
positive	59.2%	53.3%	56.0%	0.387	57.4%	52.1%	54.6%	0.434	0.773
negative	40.8%	46.7%	44.0%		42.6%	47.9%	45.4%		
HER2									
positive	25.5%	35.8%	31.2%	0.102	26.7%	37.6%	32.6%	0.088	0.758
negative	74.5%	64.2%	68.8%		73.3%	62.4%	67.4%		

PSM: propensity score matching; TNM: tumor node metastasis; ER: estrogen receptor; HER2: epidermal growth factor receptor 2; SD: standard deviation; *p* value ^α^: the probability of the hypothesis that the difference between before and after 2011 in MDT was caused by sampling error; *p* value ^β^: the probability of the hypothesis that the difference between before and after 2011 in N-MDT was caused by sampling error; *p* value ^γ^: the probability of the hypothesis that the difference between the MDT group and the N-MDT group was caused by sampling error; the *p* value comes from the *t*-test, Wilcoxon rank--sum (Mann--Whitney) test, chi-square.

**Table 4 ijerph-17-00277-t004:** The effects of MDT group and N-MDT group after PSM (patients from 2006 to 2016, *n* = 436).

Groups	1-Year Survival Rate (%)	3-Year Survival Rate (%)	5-Year Survival Rate (%)	Median Survival Time (Day)
MDT (2006–2016)	98.5	81.6	65.6	1131
N-MDT (2006–2016)	97.6	77.4	72.8	946
*p* value ^α^	0.475	0.285	0.097	0.126
Panel A				
MDT (2006–2010)	99.0	79.5	58.8	1785
N-MDT (2006–2010)	99.0	82.2	78.7	2358
*p* value ^β^	0.983	0.643	0.004	0.001
Panel B				
MDT (2011–2016)	98.1	84.1	78.8	790
N-MDT (2011–2016)	95.4	67.9	63.2	647
*p* value ^γ^	0.250	0.004	0.007	0.043

MDT: Multidisciplinary teams; N-MDT: without multidisciplinary teams’ treatment; PSM: propensity score matching; Panel A: patients who were treated during 2006–2010 after PSM; Panel B: patients who were treated during 2011–2016 after PSM; *p* value ^α^: the probability of the hypothesis that the difference between MDT (2006–2016) and N-MDT (2006–2016) was caused by sampling error; *p* value ^β^: the probability of the hypothesis that the difference between MDT (2006–2010) and N-MDT (2006–2010) was caused by sampling error; *p* value ^γ^: the probability of the hypothesis that the difference between MDT (2011–2016) and N-MDT (2011–2016) was caused by sampling error; the *p* value comes from chi-square.

**Table 5 ijerph-17-00277-t005:** The differential effects of MDT in Cox regressions.

Variables	Hazard Ratio	95% Confidence Interval	*p* Value
MDT*TIME	0.1	(0.064, 0.350)	<0.001
TNM	1.7	(1.293, 2.306)	<0.001
CCI	1.0	(0.598, 1.511)	0.830
AGE	1.0	(0.992, 1.026)	0.289
Bilateral incidence	3.0	(1.798, 5.038)	<0.001
ER	0.5	(0.312, 0.694)	<0.001
HER2	0.8	(0.499, 1.185)	0.234
TIME	2.5	(1.324, 4.558)	0.004
MDT (TIME = 0)	2.8	(1.691, 4.539)	<0.001
Panel B			
MDT (TIME = 1)	0.4	(0.201, 0.834)	0.014

MDT: Multidisciplinary teams; TNM: tumor node metastasis; CCI: Charlson comorbidity index; TIME: 1 (after2011)/0 (before 2011); MDT*TIME: the interaction effect between MDT and time node 2011, indicating the effect of well-organized MDT; Panel B: patients who were treated during 2011–2016 after PSM; *p* value: the probability of the hypothesis that the relationship between covariate and dependent variable was caused by sampling error.

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
