# Peer review of "The Improved Effects of a Multidisciplinary Team on the Survival of Breast Cancer Patients: Experiences from China"

_ijerph, 2019, doi:10.3390/ijerph17010277_

Round 1

Reviewer 1 Report

This is a much more improved manuscript. The authors have been response to the suggestions and requests made.  My only final comment is that the authors should strengthen the following sentence on study limitations: "Second, this study was limited to these several indicators. This may influence the results because of lacking sufficient covariates." I feel that the authors should list at least a couple of examples of confounders that would be valuable to have adjusted for. 

Author Response

Dear Reviewer,

Thank you very much for all your suggestions, which made the manuscript a great improvement. We value your suggestion on the second limitation of this study. We add some examples of the possible confounding factors in Line73 Page 9. "For example, the basic status of patients, tumor imaging information and tumor location are the important factors for doctors to consider the MDT treatment, which are hard to acquire in our study."

Thanks again for all your help.

Best Wishes,

Jianlong Lu

Reviewer 2 Report

I'm happy the authors have now addressed my comments. 

Author Response

Dear Reviewer,

Thank you very much for all your suggestions on the manuscript improvement, Which means a lot for us.

Thanks again.

Best wishes,

Jianlong Lu

This manuscript is a resubmission of an earlier submission. The following is a list of the peer review reports and author responses from that submission.

Round 1

Reviewer 1 Report

Abstract

The abstract does not adequately summarize the paper. For instance, the authors stated “The aim of this study was to show that a well-organized multidisciplinary team 11 (MDT) had a better effect on the survival of breast cancer patients.” MDT had a better effect compared to what?  The comparison group is also unclear in the presentation of the result:  “Cox regression model was used to measure the effect of Well-organized MDT, Disorganized MDT and the different effects between these two groups on the survival of patients. We discovered that patients who were discussed by a Well-organized MDT had a 15.64% higher survival rate at five years and had a longer survival time (HR: 0.41, 95% CI: 0.20-0.83) than those who were not discussed by an MDT.” The former sentence states that the comparison is between well-organized versus disorganized MDT but the results suggest that well-organized MDT was compared to no MDT.  Based on this, it is hard to determine if the last sentence in the abstract is accurate. 

INTRODUCTION

Their brief summary of the MDT research was lacking. For instance, I am unclear about what this sentence means: “Most of these studies thought that MDTs had a positive effect on the patients.” Do the authors mean that the “studies suggest that MDTs..”? Also, what was positively effected? The authors also write : “So, there are many differences between the Well-organized MDTs and the Disorganized MDTs.” This statement seems to come out of nowhere. It could be omitted completed. This section concludes by stating that they will “assess the different effects of an MDT during different  periods, especially after it was well-organized, on the survival of breast cancer patients”. This is not a clearly articulated objective. What is meant by “different effects?” The objective should be allow the reader to clear identify the main dependent (e.g. survival) and independent variables (e.g. MDT type)?  

METHODS          

Study Design: I think the authors mean that MDT was disorganized before (prior to) 2011 and well-organized after (post) 2011.  This is not how it was described. Also, was 2011 data included (Note that the authors tell us in the Methods section that 2012 is part of well-organized MDT)? Also the study design is an observational retrospective cohort study. Details on the propensity score matching and regression method should be moved to the data analysis (“Methods”) section.

Participants: A study flow chart would be nice. In The methods section they say that they matched, in part, on date of operation. This means that an inclusion criteria involved having surgery. This should be noted.

Intervention: There is a group of patients that received not MDT. How is it determined who received MDT and who does not?  N-MDT is not defined although the reader can deduce that the authors are referring to the patients that did not receive MDT.  

Also, Typo in Table 1 (Before2011).

Methods: This section should probably be labeled as “Data Analysis” and should go after the Variables section not before it.  The analysis plan is unclear.

RESULTS

Basic Characteristics: The description of the sample is lacking.  They should clearly describe the differences between the MDT and N-MDT samples and then briefly state that after propensity score matching- the two groups were similar.

MDT versus N-MDT: Table 4 is not clearly formatted. There is no good reason to show the analyses done on the non-matched dataset.

Effect of Well-organized MDT: The authors wrote: “As we know that construction of a Well-organized MDT was completed in 2011. Therefore, we tested the effect of a Well-organized MDT based on the survival data of patients who were treated after 2011.” This sentence should not be a part of the Results section. It would belong in the Methods section as part of the description of the data analysis plan. Does Table 5 data come from the matched dataset? This is unclear as they did a regression analysis adjusting for covariates. It is also unclear why there is attention on the covariates given the main objective of the study.

Effect of Disorganized MDT: Does Table 6 data come from the matched dataset? This is unclear as they did a regression analysis adjusting for covariates. It is also unclear why there is attention on the covariates given the main objective of the study.

Different effects between the Well-organized MDT and Disorganized MDT: The analysis is unclear to me and subsequently the results. Who is in the model? The full study sample, including bob-MDT patients? As such, it is hard to interpret the results in this section.   

DISCUSSION

I cannot fully comment on the Discussion given the questions and issues I posed in the Methods and Results section.

Reviewer 2 Report

General comments

A very interesting study, with a lot of analysis done to try and unpick the impact of multidisciplinary teams on survival of breast cancer patients. It was confusing what the authors meant by ‘well-organised MDT’, ‘disorganised MDT’ as these terms are used before they are properly defined. I also wonder if it may be better to use the term ‘impact’ rather than ‘effect’ of a MDT.

Abstract

Without knowing what ‘well-organised MDT’ and ‘disorganised MDT’ it was difficult to follow the abstract I would not say ‘and so on’ in the abstract. If there are other elements, list them. The final sentence of the abstract just repeats what is said in the sentence before. It would be more beneficial to include a statement of implications here.

Introduction

Line 36: ‘MDT meetings have been performed’ rather than ‘MDT meeting has been performed’ Be careful of making sure that the sentence makes sense after you use the acronym MDT. For example, line 46 would benefit from ‘first hospitals to start MDT meetings in 2005’ Did the studies mentioned in line 48 ‘think’ that MDTs had a positive effect on patients or did their findings show this?

Materials and methods

Need to correct definition of well organised and disorganised here – wrong way around. For all figures I would provide a footnote to remind readers of the acronym definitions Suggestion for line 76: FUSCC is one of the most ‘well-known’ rather than ‘famous’ A lot of the information in section 2.3. sounds very introductory rather than methods. I would suggest moving lines 82-92 into the introduction as that would introduce readers to MDTs in China and also provide the definitions for well-organised and disorganised MDTs. And include Table 1 following the paragraph. The manuscript is missing an analysis section.

Results

There is a need for some explanation about the changing figures included in the analyses. For example, MDT goes from n=299 in Table 2, to n=218 in Table 3. Need some explanation as to why. I would also like to know how the authors selected the matched patients from the non-MDT group when there were over 16,000 in that group. This needs to be included in the methods (analysis). All tables throughout the manuscript would benefit from ‘n’ – how many patients included in the analysis For section 3.2, does this compare the n=299 for MDT with n=16055 non-MDT? This is likely to impact the results if so. Section 3.2. need to mention that the 5 year survival is lower in MDT group after PSM The authors should state whether findings are significant rather than just saying ‘higher’ or ‘lower’ Section 3.3. first sentence not necessary, this is not a result. Section 3.3. line 25 what is the ‘better survival rate’? Need to provide some figures. Also, ‘obvious’ should not be used when referring to statistical analysis. It would be more appropriate to write ‘there were statistically significant differences in the 3- and 5- year survival’ Line 36 – this whole paragraph needs to state if the variables are beneficial for survival in the well-organised or disorganised group Overall, there are a lot of tables (8) and figures (4) and I am not sure these are all needed. Perhaps some could be included as supplementary material

Discussion

The first paragraph is background information, not discussion. This would be better suited in the introduction. Line 84: you need to be careful saying you ‘proved’ something. There were many variables at play. Perhaps ‘the study suggested that a well-organised’ would be more appropriate. There is no mention in the discussion about the 5 year survival rate in non-MDT patients being higher even after PSM. I think this is important to discuss and suggest possible reasons for this.

Round 2

Reviewer 1 Report

Abstract

Line 17: “..propensity score matching to eliminate selection bias…” PS matching can reduce or adjust for selection bias. It is an overstatement to say that it would eliminate it. I strongly suggest revising that statement on PS matching.

INTRODUCTION

Line 71-72: “…compared with patients did not accept MDT treatment or patients accepting Disorganized MDT treatment…” Were patients offered disorganized MDT and no MDT (they have to be offered in order to ‘accept’ it)? I think the authors meant that ‘compared with patients who did receive MDT treatment or received Disorganized MDT’.

METHODS          

Line 74: should be Participants NOT participants

RESULTS

Line 119: One sentence begins with a number (51% ). If sentences start with a number it should be spelled out.

The description of the sample could be better articulated as it was hard to follow. As an example, “…MDT group patients’ TNM were in stage 3 and 48% of the N-MDT were in stage 2.” This can be worded as ‘Patients in the MDT group had a higher proportion of Stage 3 tumors as compared to patients in the N-MDT group, 46% versus 13% respectively.’  Another example “Both of the MDT and N-MDT groups had a larger number of patients with single incidence, positive ER and negative HER2.” Do the authors mean that the both groups primarily had tumors that were single incidence, etc?

The Table 2 and Table 3 titles use ”PSM” which is not spelled out in either the title or a footnote. None of the Tables or Figures should have abbreviations that are not spelled out. Tables and Figures should be able to stand alone without referencing the text/narrative. Also perhaps all Table and Figures that use the PSM dataset should include a footnote. This way that do not have to keep repeating this in the narrative such as in line 2 of page 7.

Line 5 on page 7: “However, all those differences were not significant.” should read “…not statistically significant”

Line 6 on page 7: “The Kaplan-Meier survival curves shows in Figure 3…” should read “...curves shown…”

Tables 4 ,5, and 7 show the same general data but are formatted differently. Such as the label “Groups” (Table 4) versus “GROUP “(Table 5)   or the fact that ‘1-year survival rate’ is listed on one line (Table 5) versus 2 lines (Table 4). Please be consistent. Also Tables 6 and 7 could be combined into one table.

Tables 2 and 3 do not use decimal points for the %s but Tables 4 and 5 use 2 decimal points.   Please be consistent.

Line 30-31 on page 8: “Our analyses indicated that TNM stage, age, and Charlson comorbidity index had no significant influence on patients’ survival after controlling for other variables.” This sentence should be removed.  First, it is not relevant to the research question or hypothesis posed. Second, if this analysis was done on the PSM dataset then this means that these factors are already controlled for really. The statement has no real value.

Looking at Table 6, I cannot tell identify the referent group. I presume it is the N-MDT group. This should be clearly noted in the table. In fact, all the referent groups for each variable should be noted.  CCI should also be spelled out somewhere on the table. Why do hazard ratios have 6 to 7 decimal places? Standard errors have 7 decimals places and p values have 0 to 3 decimal places.  Please reduce the number of decimal places (at least for the hazard ratios and SEs) and be consistent. Also Table 6 presents the analysis for disorganized MDT first then well-organized MDT results. Yet, the narrative first speaks of the well-organized MDT results. The results of both can be summarized in one paragraph. Reference to any of the other covariates is not necessary as they not speak to the research question or hypothesis.

Table 8 is difficult to interpret. I am not sure, but a statistician should confirm this—shouldn’t the model include TIME in order to make the model hierarchically valid? It is used in the interaction terms yet it is not included as a main effect. Also, they write : “…was 0.34, meaning the Well-organized MDT had a positive effect on the survival compared to the Disorganized MDT group.” First, the sentence should include the HRs and 95% CIs for both Well-organized MDT and Disorganized MDT. The reference group is the N-MDT group. That is not clear on the table or the interpretation of the HRs.  Note my earlier comment about all the decimal places.

DISCUSSION

The first 2 sentences should leave out the findings regarding the covariates. Also, the data suggests that disorganized MDT may have been harmful. That finding must be more tempered. Especially in light of the study’s limitations. Also, they report: “Our results indicated that the Well-organized MDT is more effective than the Disorganized MDT on improving the survival of patients.” This is not correct because the reference was always the N-MDT group.  Discussion of the covariate findings should be eliminated.

The authors note 3 strengths but their first and second stated strength are similar. The limitations should include the selection bias—a huge proportion of their sample was culled due to missing data. It is unclear how that would bias the observed results. It is not clearly stated but the other major limitation is the residual confounding due to the lack of sufficient covariates.  

CONCLUSIONS

What do the authors mean by “there was no substantial survival difference 1 between the N-MDT and Disorganized MDT…”?  There WAS a difference I survival between the two latter groups. Albeit I would not high light this.  They should also articulate how future studies can be improved to better inform this area of study.

Reviewer 2 Report

Thank you to the authors for making substantial improvements to the manuscript. I have some further comments which I believe will improve the clarity of the manuscript. 

I still think there needs to be some explanation of well organised and disorganised in the abstract

Need to make clear non-MDT is different to disorganised – so MDT patients include those in well organised and disorganised groups

Line 70-72: Can’t use the terminology ‘accepting’ MDT or not as I can’t see that patients would have had a say whether they did or not?

Significant differences between MDT and non-MDT so need to account for these

All tables need footnotes for anagrams

Page 4, line 82: ‘If they had not received an effective MDT, the 5-year survival rate would be lower because of severity of disease’ You cannot say this with certainty, it can be suggested
